Comparative proteomic analysis between mature and germinating seeds in Paris polyphylla var. yunnanensis

Ling Li-Zhen 121302168@qq.com
Zhang Shu-Dong
Key Laboratory of Liupanshui City for Study and Utilization of Ethnic Medicinal Plant Resources of Western Guizhou Province, Liupanshui Normal University , Liupanshui , China
Uversky Vladimir
Electronic publication date: 2022 May 11
Publication date: 2022
Volume: 10
Electronic Location ID: e13304
Received 2021 Dec 16; Accepted 2022 Mar 29
Copyright: © 2022 Ling and Zhang
Copyright year: 2022
Copyright holder: Ling and Zhang
License: This is an open access article distributed under the terms of the Creative Commons Attribution License, which permits unrestricted use, distribution, reproduction and adaptation in any medium and for any purpose provided that it is properly attributed. For attribution, the original author(s), title, publication source (PeerJ) and either DOI or URL of the article must be cited.
License URL: https://creativecommons.org/licenses/by/4.0/

Keywords: Proteomics, Seed germination, Abscisic acid, Gibberellins, Paris polyphylla var. yunnanensis

Funding: Science and Technology Innovation Platform and Talent Team Construction Project of Liupanshui 52020-2019-05-05 Natural Science Research Project of the Education Department of Guizhou Province QJHKYZ[2018]067 This work was supported by the Science and Technology Innovation Platform and Talent Team Construction Project of Liupanshui (No. 52020-2019-05-05) and Natural Science Research Project of the Education Department of Guizhou Province (No. QJHKYZ[2018]067). The funders had no role in study design, data collection and analysis, decision to publish, or preparation of the manuscript.

==============================
The long dormancy period of Paris polyphylla var. yunnanensis seeds affects the supply of this scarce plant, which is used as an important traditional Chinese medicine. Mature seeds with a globular embryo and germinating seeds with developed embryo were used to explore the mechanisms of seed germination in this species. The protein profiles between the mature and germinating seeds were compared using the isobaric tags for relative and absolute quantification (iTRAQ) approach. Of the 4,488 proteins identified, a total of 1,305 differentially expressed proteins (DEPs) were detected. A Kyoto Encyclopedia of Genes and Genomes (KEGG) analysis of these DEPs indicated that metabolic pathways and the biosynthesis of secondary metabolites were the two top pathways. Additionally, phytohormone quantification shows that the abscisic acid (ABA) level significantly decreased, whereas the GA3 level dramatically increased among nine endogenous gibberellins (GAs), resulting in a significant increase of the GA3/ABA ratio in germinating seeds. The biosynthesis pathways of carotenoid as a precursor for ABA production and GA were further analyzed, and showed that proteinic expressions of the candidate genes in the two pathways did not correlate with the transcriptional abundances. However, 9-cis-epoxycarotenoid dioxygenase (NCED), a rate limited enzyme for ABA biosynthesis, was significantly decreased in mRNA levels in germinating seeds. By contrast, gibberellin 20-oxidase (GA20ox), a key enzyme GA biosynthesis, exhibited the major increase in one copy and a slight decrease in three others at the protentional level in germinating seeds. Gibberellin 2-oxidase (GA2ox), an inactivate enzyme in bioactive GAs, has the tendency to down-regulate in mRNA or at the proteinic level in germinating seeds. Altogether, these results suggested that the analyses of ABA and GA levels, the GA3/ABA ratio, and the expressional patterns of their regulatory genes may provide a novel mechanistic understanding of how phytohormones regulate seed germination in P. polyphylla var. yunnanensis.

Introduction

The genus of Paris (Melanthiaceae) consists of more than 24 species, the majority of which are distributed in China (Li, 1984). Paris polyphylla var. yunnanensis is used in traditional Chinese medicines and is commonly known as “Chonglou” in China (Li, 1984). The root of this plant has been reported to possess many bioactive compounds. Steroidal saponins are among the main bioactive ingredients, and have hemostatic, analgesic, antimicrobial, and anti-inflammatory effects (He et al., 2006; Liu et al., 2015; Wang et al., 2015; Zhang et al., 2004). The wild plant has become endangered due to unrestrained exploitation for medicinal purposes. A slow germination rate and long dormancy period have also contributed to the endangered status of this plant. Approximately 40% of Chonglou seeds can germinate after experiencing the natural dormancy period of more than 18 months under natural field conditions (Li, 1982). Presently, the wild plants cannot meet the seed demands for pharmaceutical use and the cultivation of Chonglou at the field level is necessary. Priming pretreatments have succeeded in accelerating seed germination and improving seedling uniformity by subjecting seeds to fluctuating temperatures and growth regulators (i.e., gibberellins (GAs)) (Chen et al., 2011; Zhang, Ma & Hu, 2018; Zhang et al., 2013; Zhang & Ling, 2019). For example, the germination rate of seeds can reach 80% with a treatment of 200 ppm GA3, as outlined in our previous method (Zhang & Ling, 2019).

Seed dormancy is an innate seed property and is divided into five classes based on the morphological and physiological properties of the seed: physiological (PD), morphological (MD), morphophysiological (MPD), physical (PY), and combinational (PY+PD) (Baskin & Baskin, 2004). Of these, MPD is considered to be the most ancestral type within the angiosperms (Baskin & Baskin, 2004). Seeds with MPD display a combination of MD (seeds are characterized by underdeveloped embryos that may or may not be differentiated) and PD (a physiological mechanism prevents elongation of the axis, usually observed as radicle protrusion). Studies have demonstrated that the seed embryo is at the globular stage when the fruits of Chonglou are mature (Chen et al., 2015; Huang et al., 2008). The seeds of Chonglou must experience the morphological post-maturation and physiological post-maturation in the process of germination, which is typical of MPD. Evidence has shown that the seed embryo development in Chonglou is in association with reduced abscisic acid (ABA) and increased GA levels (Chen et al., 2015; Huang et al., 2008). Furthermore, ABA and GA related genes (CYP707A, NCED, GA20ox2, GA20ox3, ABI2, PP2C) were shown to exhibit different transcriptional levels during embryo development. Six of the GAox genes involved in the GA metabolic pathway have been cloned (Qi et al., 2013; Zhang et al., 2017). The miRNAs in the germinating seeds have been identified and their targets were found to be enriched during cellular, metabolic, and genetic information processing (Zhang & Ling, 2019). The miR159 a/b was predicted to target the members of MYB transcription factor family (Zhang & Ling, 2019). This miRNA regulatory module has been shown to be the positive regulator of ABA signaling and affects the GA-mediated programmed cell death during seed germination in other plants (Alonso-Peral et al., 2010; Reyes & Chua, 2007). These studies have mined candidate genes for clarifying the basis of seed germination in Chonglou at the different levels.

The use of proteomics has expanded our ability to monitor and analyze the spatial and temporal properties of proteins. This approach can bring robust information about the relationship between biological functions and physiological changes. In the past two decades, two-dimensional electrophoresis (2-DE), a gel-based technique, has been widely used for analyzing the protein compositions in Arabidopsis (Gallardo et al., 2001), maize (Guo et al., 2013), rice (He & Yang, 2013) and Norway maple (Pawlowski, 2009) during seed germination. In maize, the proteins involved in germination-related hormone (ABA and GA) signal transduction networks have been analyzed during the early stages of seed germination (Guo et al., 2013). Isobaric tags for relative and absolute quantification (iTRAQ) is a chemical labeling method used in quantitative proteomics by the tandem mass spectrometry (MS/MS) (Ross et al., 2004). Due to the high sensitivity and accuracy, iTRAQ-based proteomics has been widely applied in seed development and germination (Peng et al., 2018; Zhang et al., 2018; Zhu et al., 2018). However, the protein profiles of seed germination in Chonglou has not been investigated until now. To better understand the seed gemination mechanism in this species, the proteomic profiles were detected using the iTRAQ method. The levels of ABA and endogenous GAs in the mature seeds and the germinating seeds at the 20th day of incubation in Chonglou were detected and the candidate genes responsible for ABA and GA biosynthesis were analyzed. Finally, we analyzed the relationship between the differentially expressed mRNAs and proteins of these candidate genes in mature and germinating seeds in Chonglou.

Materials and Methods

Plant materials

The seedlings of Chonglou were planted in Ludian, Yunnan Province, southwest China. The seeds were collected after the fruits matured. The collected mature seeds were primed with 200 ppm GA3 and were kept at 20 °C to germinate, according to our previously described method (Zhang & Ling, 2019). The priming resulted in the morphological post-maturation of the globular seed embryo on the 20th day of seed incubation (Zhang & Ling, 2019). The mature seeds were collected in the field and the seeds that were germinated until the 20th day of incubation were used as experimental materials. All materials were immediately snap-frozen in liquid nitrogen and stored at −80 °C until used. Three biological replicates of these experimental materials were used to detect the proteomics, the transcriptome and the levels of GAs and ABA, respectively.

The measurement of plant hormones

In this study, two major plant hormones, ABA and GAs, were measured in mature and germinating seeds, respectively. For ABA, the extraction and quantification were carried out using an HPLC-MS/MS (LCMS-8040 System; Shimadzu, Somerset, NJ, USA) (Thameur, Ferchichi & López-Carbonell, 2011). The determination of GAs were performed on an HPLC-MS/MS (LCMS-8040 system; Shimadzu, Somerset, NJ, USA) (Kojima & Sakakibara, 2012).

Protein extraction, digestion, and iTRAQ labeling

Protein samples were prepared as previously described (Yang et al., 2019) and the detailed protocol of protein extraction was shown in Fig. S1. Each sample was ground in liquid nitrogen with the pallet and then mixed with 1.5 ml lysis buffer containing 7 M urea, 2 M thiourea, 0.1% CHAPS and a protease inhibitor. The supernatant was collected after centrifugation at 25,000g for 20 min at 4 °C and then precipitated by adding five times the sample volume of 10% (w/v) trichloroacetic acid/acetone (−20 °C) to the tube overnight. The samples were centrifugated at 15,000g for 20 min at 4 °C and the pellet was rinsed three times with pre-cooled acetone. The precipitate was air-dried and dissolved in lysis buffer, which was used to measure the protein concentration with the Bradford assay.

The 100 μg protein aliquots were digested with 2.5 μg of trypsin at 37 °C overnight. The digested peptides were desalted with Strata X column and then vacuum-dried. Approximately 100 μg of desalted peptides were labelled with isobaric tags from the iTRAQ Reagent-8plex Multiple Kit according to the manufacturer’s instructions. The labelled peptides were pooled in equal amounts and dried by vacuum centrifugation.

LC-MS/MS analysis

The labelled peptide mixtures were redissolved in solution A (5% ACN and 95% ddH2O, pH 9.8) and then fractionated with a Gemini C18 column in an LC-20AB HPLC system (Shimadzu, Kyoto, Japan). Next, retained peptides were eluted with solution B (95% ACN and 5% ddH2O, pH 9.8). After separation, the collected fractions were concentrated by vacuum centrifugation and resuspended in solution C (2% ACN, 0.1% formic acid). The fractions were analyzed with an LC-20AD HPLC system (Shimadzu, Kyoto, Japan) coupled to a Q-Exactive mass spectrometer (Thermo Fisher Scientific, San Jose, CA, USA). The Q-Exactive mass spectrometer was operated in a data-dependent acquisition mode to switch automatically between MS and MS/MS acquisitions. The survey of the full-scan MS spectra (m/z 350-1,500) was set for 30 s dynamic exclusion. The mass spectrometry proteomics data have deposited to the Zenodo data set (DOI 10.5281/zenodo.6370960) and available.

The bioinformatics analyses of protein data

The raw mass data were used to search against the ‘non-redundant protein sequences (nr)’ database by the blast program. Protein identification was performed using the Mascot sever (Version 2.3.02) against the transcriptome data published in our previous study (Ling et al., 2017; Zhang & Ling, 2019). The confident protein identification involved at least one matched unique peptide with a confidence interval ≥95% according to the mascot probability scores. Meanwhile, iQuant software (v2.2.1, BGI-Shenzhen, Guangdong, China) (Wen et al., 2014) was used to quantify proteins in this study. The false discovery rate (FDR) analyses were conducted using the pocked protein FDR strategy (Savitski et al., 2015). To ensure the high confidence, the peptides with a global FDR_?% were removed in the protein analysis. The significant differentially expressed proteins (DEPs) were selected by fold-change ratio (>1.5 or <0.67, p < 0.05). Functional category analyses were performed with Blast2GO and the cluster of orthologous groups of proteins (COG). The metabolic pathway of DEPs was predicted using the Kyoto Encyclopedia of Genes and Genomes (KEGG) (http://www.genome.jp/kegg/) database. A p-value ≤ 0.05 was used as the threshold of significant enrichment of GO and KEGG pathways.

Transcriptome analysis

Transcriptome data of mature and germinating seeds were assembled and annotated in our previous study (Ling et al., 2017; Zhang & Ling, 2019). Each sample had three biological replicates. The transcriptome sequencing data for mature and germinating seeds used in this study are available in the NCBI SRA database under accession numbers SRX2335704 and SRP126881, respectively. The mRNA abundance was estimated using the number of uniquely mapped fragments per kilobase of transcript per million mapped reads (FPKM) method.

Differentially expressed genes (DEGs) were detected using NOIseq with a threshold of 0.8 (Tarazona et al., 2011). Genes were regarded as significantly differentially expressed when a p-value < 0.001, a FDR < 0.01 and an absolute log2 ratio > 1 in sequence counts across sample transcriptomes were achieved. The FDR was calculated to correct the p-value; the smaller the FDR, the smaller the error in judgment of the p-value. The p-value was calculated as follows:

p=1−∑i=0m−1(Mi)(N−Mn−i)(Nn)

where m indicates the DEG number in each GO/KO term; M indicates all gene number in each GO/KO term; N represents the number of all transcripts with GO/KO annotation. The corrected p-value ≤ 0.01 as the threshold determined the significant enrichment of the gene sets.

Results

iTRAQ analyses of proteins in mature and germinating seeds

In total, 66,482 spectra were generated from mature and germinating seeds of the 20th day based on the iTRAQ-LC-MS/MS proteomic analyses (Table 1). After filtering out low-scoring spectra, 43,762 unique spectra were matched to 4,488 proteins at the given thresholds (FDR < 1%, 95% confidence, at least one unique peptide in each positive protein identification) (Table 1). The length of these identified proteins ranged from 34 to 5,078 AA, of which proteins of less than 1,000 AA were the most abundant (94.81%) (Table S1). The average length of the proteins was 441 AA and a few of proteins (0.47%) were greater than 2,000 AA in length (Table S1). Among the identified proteins in mature and germinating seeds, the proteins with a relative abundance of >1.5-fold or <0.67-fold were considered to be DEPs. In this study, a total of 1,305 (642 down-regulated and 663 up-regulated) DEPs were identified in germinating seeds, when compared with mature seeds (Table 1, Table S2 and Fig. S2).

Table 1 Statistics for the proteins and differentially expressed proteins.

Total spectra	Spectra	Unique spectra	Peptide	Unique peptide	Protein	DEPs	
259,124	66,482	43,762	18,136	14,320	4,488	1,305	

Functional annotation of the DEPs

To understand the functions of the identified DEPs, COG, GO and KEGG enrichment analysis were performed. COG analysis showed that these DEPs were grouped into 24 functionally categories. Most of the DEPs were clustered into the “Carbohydrate transport and metabolism” category, followed by “Posttranslational modification, protein turnover, chaperones” and “General functional prediction only” (Fig. 1). In addition, “Energy production and conversion”, “Amino acid transport and metabolism” and “Lipid transport and metabolism” clusters were found to be the other abundant categories (Fig. 1). The GO enrichment analysis revealed that the DEPs were associated with the different functional processes. At the molecular level, the majority of DEPs were involved in “catalytic activity” and “binding activity” (Fig. 2). Meanwhile, the DEPs were linked to “cell”, “cell part”, “organelles” and “membrane” at the cellular component level (Fig. 2). Most of the DEPs were involved in “cellular and metabolic processes” in biological process. These results indicated that the DEPs were involved in the different metabolic and cellular processes and may localize in the different cell parts and organelles.

Figure 1 Histogram of COG classification of DEPs.

Figure 2 GO categories assigned to DEPs.

Pathway-based analysis is helpful to further understand the biological functions and gene interactions. KEGG pathway analysis revealed that the DEPs were enriched in 132 pathways at the third level. Of these, “metabolic pathways” (359), “biosynthesis of secondary metabolites” (237), “carbon metabolism” (56), “starch and sucrose metabolism” (50), “protein processing in endoplasmic reticulum” (47), “biosynthesis of amino acids” (40), and “phenylpropanoid biosynthesis” (37) were found to be the top seven annotated KEGG pathways for the DEPs (Fig. 3A). At the second level, “metabolic pathways”, “biosynthesis of secondary metabolites” and “carbon metabolism” were classified into “Global and overview maps”, which accounted for the largest proportion (Fig. 3B). However, the DEPs in “metabolic pathways” and “biosynthesis of secondary metabolites” had a small enrichment factor (<0.4) although they had a large enrichment number (Figs. 3A and 4). This data meant that no less than 40% proteins in these above-mentioned pathways showed the different expression patterns between two samples. By contrast, “anthocyanin biosynthesis” exhibited a largest value of rich factor (= 0.8), followed by “photosynthesis”, “folate biosynthesis” and “flavonoid biosynthesis” (Fig. 4).

Figure 3 The top seven KEGG pathways of DEPs at the third level (A) and the KEGG pathways at the second level (B).

Figure 4 Enrichment of DEPs in KEGG pathways.

The comparison of ABA and GAs levels between mature and germinating seeds in Chonglou

Previous studies have revealed that ABA and GA play important roles in seed germination in Chonglou (Chen et al., 2011; Meng et al., 2006; Zhao, Luo & Li, 2014). Figure 5A shows that the average ABA content of mature seeds was 55 ng/g FW, whereas that of germinating seeds of the 20th day was significantly decreased to 9.43 ng/g FW (p < 0.05). In this study, nine endogenous GAs (GA1, GA3, GA4, GA7, GA9, GA12, GA19, GA24 and GA44) were unequivocally quantified in mature and germinating seeds of the 20th day. Our results indicated high amounts of GA1 and GA3 in the mature and germinating seeds. The content of GA3 was 179.62 ng/g FW in germinating seeds, and significantly higher than that of mature seeds (p < 0.05) (Table 2). Although the content of GA1 showed the similar change pattern as that of GA3 and no significant increase occurred in GA1 (Table 2). The levels of other GAs (GA4, GA7, GA9, GA12, GA19, GA24 and GA44) showed low amounts in mature and germinating seeds. For example, GA4 and GA12 were not detected in mature seeds, whereas GA7, GA9 and GA24 were not detected in germinating seeds of the 20th day (Table 2). We found that GA7 showed the significant reduction during seed germination. Altogether, the concentration of the GA3 and GA7 showed the significant and opposite changes during seed germination. Meanwhile, we separately calculated the ratio of GA3/ABA and GA7/ABA and found they were similar to the respective GAs levels (Fig. 5B).

Figure 5 The comparison of ABA concentration (A) and the ratios of GA3/ABA and GA7/ABA (B) in mature and germinating seeds.

Note: *Difference is significant at the 0.05.

Table 2 The comparison of the GAs levels between mature and germinating seeds.

Samples	Levels of hormones (ng/g FW)	
GA1	GA3	GA4	GA7	GA9	GA12	GA19	GA24	GA44	
Mature seed	21.27 ± 1.9	40.69 ± 4.9	0 ± 0	0.75 ± 0.16	0.25 ± 0.06	0 ± 0	0.69 ± 0.31	1.14 ± 0.17	4.91 ± 0.54	
Germinating seed	48.79 ± 0.72	179.62 ± 31*	1.73 ± 0.17	0 ± 0*	0 ± 0	0.22 ± 0.03	1.42 ± 0.12	0 ± 0	2.74 ± 0.42	
Note:

* The average difference is significant at the 0.05 level.

Characterization and expressional analyses of carotenoid catabolic pathway genes

In plants, the carotenoids biosynthetic pathway provides the precursor for ABA synthesis (Finkelstein, 2013). To understand the molecular basis of ABA level changes during seed germination, we identified the candidate genes involving in carotenoid biosynthetic/catabolic pathway. In this study, a total of 29 candidate genes encoding 26 enzymes were found and their detailed information were summarized in Table 3 and Fig. 6. Of these candidate genes, only six enzymes, including carlactone synthase/all-trans-10′-apo-beta-carotenal 13,14-cleaving dioxygenase (CCD8), fabG, lycopene epsilon-cyclase (LCYE), peroxidase, VPS54 and UDP-glucose:glucosyl (waaD) had only one copy, whereas the rest of carotenoid genes had more than two copies (Table 3 and Table S3).

Figure 6 The ABA biosynthesis pathway.

The enzymes (the font)/transcripts (arrow in the bracket) with down-regulated expression level are shown in blue; the enzymes (the font)/transcripts (in the bracket) with up-regulated expression level are shown in red; the enzymes (the font)/transcripts (in the bracket) with no change are shown in black. The asterisks (*) identify enzymes encoded by several gene isoforms.

Table 3 Candidate genes involved in carotenoid biosynthesis pathway and their expression changes.

Enzyme	EC	Number of unigenes	mRNA level	Protein level	
AAO1_2	1.2.3.7	4			
AAO3	1.2.3.14	10	up	down	
ABA2, SDR	1.1.1.288	14	up		
AOG	2.4.1.263	2			
BHY	1.14.13.129	6		down	
CCD7	1.13.11.68	5			
CCD8	1.13.11.691.13.11.70	1			
crtB	2.5.1.32	12		down	
CRTISO	5.2.1.13	6		down	
CYP707A	1.14.13.93	12	up/down		
DWARF27	5.2.1.14	11	up		
fabG, OAR1	1.1.1.100	1			
FMN2		2	up		
FUCA	3.2.1.51	5	up		
LCYB, crtL1, crtY	5.5.1.19	2			
LCYE, crtL2	5.5.1.18	1			
LUT1, CYP97C1	1.14.99.45	2			
LUT5, CYP97A3	1.14.-.-	10	up		
NCED	1.13.11.51	4	down		
PAF1		2	up	up	
PDS, crtP	1.3.5.5	5	up		
peroxidase	1.11.1.7	1			
TR1	1.1.1.206	2	up		
VDE, NPQ1	1.23.5.1	3	up/down	down	
VPS54		1	up		
waaD	2.4.1.-	1			
ZDS, crtQ	1.3.5.6	4	up/down	down	
ZEP, ABA1	1.14.13.90	18		up	
Z-ISO	5.2.1.12	2			

When germinating seeds were compared with mature seeds, 14 enzymes exhibited the different expressional patterns at the transcriptional level (Table 3 and Table S3). The mRNA levels of abscisic-aldehyde oxidase (AAO3), xanthoxin dehydrogenase (ABA2), beta-carotene isomerase (DWARF27), FMN2, alpha-L-fucosidase (FUCA), CYP97A3, PAF1, 15-cis-phytoene desaturase (PDS), tropinone reductase I (TR1) and VPS54 in germinating seeds were significantly higher than those in the mature seeds. In contrast, 9-cis-epoxycarotenoid dioxygenase (NCED) which is a rate limited enzyme responsible for ABA biosynthesis showed a significant decrease in the transcriptional level in germinating seeds. In addition, 11 transcripts encoding three enzymes, zeta-carotene desaturase (ZDS), zeaxanthin epoxidas (ZEP) and CYP707A were observed as up-regulated or down-regulated significantly (Table S3), which indicated that there are multiple ZDSs as well as ZEPs and CYP707As involved in ABA biosynthesis.

The pattern changes of the carotenoid candidate genes were analyzed at the translational level. Our results indicated that a total of eight candidate genes, including AAO3, beta-carotene 3-hydroxylase (BHY), 15-cis-phytoene synthase (crtB), carotenoid isomerase (CRTISO), violaxanthin de-epoxidase (VDE), ZDS, PAF1 and ZEP (Table 3, Table S3 and Fig. 6) were significantly changed. Of them, only the latter two genes were significantly up-regulated at the translational level during seed germination (Table 3, Table S3 and Fig. 6). In principle, the more mRNA molecules that are present in the cell, the more proteins can be synthesized. Therefore, the correlation of between mRNA and protein expression profiles of the carotenoid candidate genes was analyzed. However, we found that almost none of the unigenes were simultaneously and significantly changed at the translational and transcriptional levels (Table S3). There was one exception: one unigene encoded ZDS was significantly down-regulated at the mRNA and protein levels (Table S3 and Fig. 6). These results indicated that the mRNA and protein changes of the carotenoid candidate genes were not correlated.

Expressional analyses of GA biosynthesis genes at the different levels

We examined the expression of the key genes involving in GA biosynthesis in mature and germinating seeds at the different levels. The proteomic data analysis indicated that the enzymes of gibberellin 20-oxidase (GA20ox) and gibberellin 2-oxidase (GA2ox) exhibited a significant difference (Table S4). Of the three unigenes of GA20ox, two demonstrated slight decreases whereas one increased 3.6 times increase in germinating seeds compared to mature seeds (Table S4). By contrast, one GA2ox unigene exhibited a significant decrease in the germinating seeds (Table S4). Transcriptomic data analysis showed that only GA20ox and GA2ox had significantly changed patterns. Two copies of GA20ox had contrary expressional patterns and three copies of GA2ox were down-regulated during seed germination (Table S4). However, the GA20ox or GA2ox copies did not simultaneously show the significant changes at transcriptional and translational levels and there was still no correlation between the transcriptional and translational levels in GA20ox or GA2ox, two key enzymes involving in GA metabolic pathway (Table S4).

Discussion

The mature seed of Chonglou contain the globular stage embryo and possess the typical characteristics of MPD. Exogenous GA3 treatment disrupts the period of seed dormancy and promotes the formation of a mature embryo on the 20th day of incubation (Zhang & Ling, 2019). Two sets of seeds used as the experimental materials provided an opportunity to study the seed dormancy and germination mechanism in this species. GAs are a group of tetracyclic diterpenes and a total of 136 gibberellins have been identified, but only few (GA1, GA3, GA4 and GA7) are biologically active (Yamaguchi, 2008). The accumulation of these bioactive GAs is generally thought to contribute to seed germination (Yamaguchi, 2008; Wang et al., 2018). In this study, we found that the above-mentioned four biologically active GAs (GA1, GA3, GA4 and GA7) were present in the seeds of Chonglou. However, only GA3 increased in the process of seed germination (Table 2). We thus inferred that GA3 may be the major bioactive GAs involving in the seed germination of Chonglou.

ABA has the antagonistic effect on bioactive GAs and greatly declines during seed germination of Chonglou (Fig. 5A). However, an increasing amount of evidence has revealed that the balance between ABA and GA signaling produces germination potential, rather than one or the other alone (Liu et al., 2014; Penfield & King, 2009; White, Proebsting & Rivin, 2000). A low GA/ABA ratio in early seed development programming is critical for the germination suppression and maturation induction (White, Proebsting & Rivin, 2000). Previous studies demonstrated that the changes in the GA/ABA ratio have a higher positive correlation with embryo growth than the levels of ABA or GA (Chen et al., 2011; Pu et al., 2016). Therefore, the GA/ABA ratio may affect the seed germination in Chonglou more significantly. The impact of a high ratio of GA3/ABA in germinating seeds may be 25 times greater than that in dormant mature seeds (Fig. 5B). Moreover, fluctuating temperature stratification has been shown to effectively break the period of seed dormancy but the globular embryo requires approximately 40 days to complete its development, in contrast to the 20 days required by the exogenous application of GA3 (Chen et al., 2011; Pu et al., 2016; Zhang & Ling, 2019). Therefore, the two priming methods reveal the different speeds of germination in Chonglou. Studies have shown that only the GA/ABA ratio beyond a certain threshold result in seed germination (Liu et al., 2014; Sgamma, 2017). We found that the GA/ABA ratio by GA3 treatment is sixtyfold when compared with that by fluctuating temperature stratification when the undeveloped embryo completes its morphology. Therefore, we inferred that the exogenous application of GA3 can rapidly promote an increase of the GA3/ABA ratio to the threshold and effectively reduce the seed germination time.

We also characterized 26 enzymes involved in the carotenoid biosynthesis pathway (Table 3 and Fig. 6), which were responsible for ABA biosynthesis. Among these enzymes, we found that they exhibited the different expression patterns at the transcriptional and translational levels. The unigenes encoding 11 enzymes exhibited significant expression patterns at the transcriptional level but these were absent in changes of protein expression (Tables 3 and Table S3). These differences may be explained by the strict regulation of protein synthesis in multiple steps comprising transcription, translation, post-translational processing and modification, and finally the synthesis of the mature protein. The majority of the carotenoid biosynthesis genes exhibited the up-regulated expressional patterns at the transcriptional level (Tables 3 and Table S3). By contrast, NCED, a rate limited enzyme responsible for ABA biosynthesis was down-regulated at the transcriptional level and was not detected in translational level (Tables 3 and Table S3 and Fig. 6). This may contribute to the decrease of ABA levels during seed germination in Chonglou.

In the GA metabolic pathway, GA20ox and gibberellin 3-oxidase (GA3ox), catalyze gibberellin intermediates to the bioactive forms, while GA2ox deactivates the bioactive GAs (Yamaguchi, 2008). Three enzymes were studied, however only GA20ox and GA2ox showed significant changes at the transcriptional and translational levels (Table S4). For GA20ox, we found that there were more than two copies and each had a different expressional pattern. Therefore, we inferred that the Chonglou genome contains more than two paralogous genes of GA20ox which may be multifunctional. In fact, this enzyme was proven to be a multigene family in many plant species, such as Arabidopsis and rice (Yamaguchi, 2008). The enzyme of GA2ox was found to act similarly in this study and four GA2ox genes have been cloned in Chonglou (Zhang et al., 2017). Additionally, some copies of GA20ox were down-regulated at the transcriptional or translational level. A previous study demonstrated that three GA20ox genes were dramatically reduced after the application of bioactive GA under feedback control in Arabidopsis (Phillips et al., 1995). Therefore, the decreased expressional patterns of GA20ox genes may be affected by the application of GA3 and the up-regulation of several GA20ox genes and the decrease of several GA2ox genes may lead to the increase of bioactive GAs. However, the functions of the different GA20ox and GA2ox genes in embryo development and seed germination in Chonglou require further study.

Conclusions

In this study, the proteinic profiles and the levels of phytohormone ABA and GAs were compared between mature and germinating seeds in Chonglou. Our results indicated that a total of 1,305 DEPs were detected and were mainly enriched in metabolic pathways. The results indicated that GA3 and ABA participate in the process of seed germination, and the GA3/ABA ratio may have a greater effect on seed germination. Proteomics data were used to further analyze the expression patterns of candidate genes involving in the biosynthesis of two phytohormones in combination with the transcriptome data (Ling et al., 2017; Zhang & Ling, 2019). Our results demonstrated that the descending NCED transcripts may contribute to the decrease of ABA levels, whereas both the up-regulation of GA20ox and the down-regulation of GA2ox are beneficial for the accumulation of bioactive GAs during seed germination. This study provides a theoretical foundation for the further exploration of the germination mechanism by the function analysis of the regulatory genes in Chonglou.

Supplemental Information

Supplemental Information 1 Supplemental Figures and Tables.

Figure S1 The flowchart of protein extraction protocol.

Figure S2 Volcano plot of protein expression changes. Note: Red and green dots represent protein expressed at significantly higher (n = 663) or lower (n = 642) levels during germination, respectively. S20 indicates the germinating seeds at the 20th day of incubation.

Table S1 The size distribution of all identified proteins.

Table S2 All DEPs detected via Mass Spectrometry.

Table S3 Genes involved in carotenoid biosynthesis pathway and their expressions at the different level.

Table S4 Genes involved in GA biosynthesis pathway and their expressions at the different levels.

Click here for additional data file.

We thank Xu Kun from the Kunming Institute of Botany, Chinese Academy of Sciences for the identification of Chonglou in this study.

Additional Information and Declarations

Competing Interests

Author Contributions

DNA Deposition

Data Availability

The authors declare that they have no competing interests.

Li-Zhen Ling conceived and designed the experiments, analyzed the data, prepared figures and/or tables, authored or reviewed drafts of the paper, and approved the final draft.

Shu-Dong Zhang performed the experiments, analyzed the data, prepared figures and/or tables, authored or reviewed drafts of the paper, and approved the final draft.

The following information was supplied regarding the deposition of DNA sequences:

The transcriptome sequencing data for seed and germinating seeds are available at NCBI SRA: SRX2335704 and SRP126881, respectively.

The following information was supplied regarding data availability:

The raw measurements are available in the Supplemental Files.

The mass spectrometry proteomics data are available at Zenodo: Lizhenling, & Shudongzhang. (2022). The proteomics data of Paris polyphylla var. yunnanensis [Data set]. Zenodo. https://doi.org/10.5281/zenodo.6370960.

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
