# Peer review of "Comparative proteomic analysis between mature and germinating seeds in Paris polyphylla var. yunnanensis"

_PeerJ, doi:10.7717/peerj.13304_

## Round 0.1 · original submission · Major Revisions

Please address the concerns of both reviewers and revise the manuscript accordingly.

·

Basic reporting

The balance between ABA and GA is crucial for seed dormancy. Zhang and Li showed us that the levels of several GAs were significantly increased in germinating Chonglou seeds, whereas the amount of ABA was sharply reduced. Through proteomic and transcriptome analysis, the authors have pointed out that the ABA synthetic related enzymes and their responsible genes have altered in germinating Chonglou seeds. The results were consistent with previously study. However, the article is not well written, and some results need to be discussed.

Experimental design

The metods are detail described. Experimental design should be completion: the transcriptome should be verified by real-time quantitative reverse transcription-PCR.

Validity of the findings

The conclusions is appropriately.

Additional comments

Comments to author,
I have some concerns.
Major concerns:
1. The English language must be improved to ensure that an international audience can clearly understand your text.
Examples: Line 42, this species; Line 93, germinating; Line19-20, 44-45, I could not get what the authors wanted to say. There are too many such incomprehensible sentences in the whole manuscript.
2. How about the GA synthetic enzymes in germinating Chonglou seeds
3. The transcriptome should be verified by real-time quantitative reverse transcription-PCR, and also the expression level of GA synthetic genes.
Minor concerns:
1. Paris polyphylla var. yunnanensis is known as Chonglou in China. It is better using Latin and Chinese names in the beginning and then only using Chonglou.
2. Figures should be improved:
The resolution of all figures is poor;
It is better to show the data in descending order for Figures 1 and 2;
Figure legends should be detail.
3. Through Table 2, I see that the levels of serval GAs was changed. However, the authors only GA3 and GA7 were significantly increased, how about the orthers?
4. Also in Table 2, statistical analysis and significance labeling should be presented conventionally.
5. The discussion is poorly written. The state of seed dormancy is correlated with ABA/GA ratio (White et al., 2000; Penfield and King, 2009; Liu et al., 2014). The authors should carefully read these articles and fully discussed the relevance between ABA/GA ratio and the germination of Chonglou.

Reviewer 2 ·

Basic reporting

In the discussion part, I suggest the authors discuss the significance of their findings, for example, how their findings will benefit the seed germination in the field.

Even though it is a supplementary figure, figure S1 doesn’t seem very necessary. It looks like only part of the figure S2 is shown in the file. More information such as the meaning of green/red is needed. In figure 4, the dot of “metabolic pathways” is not fully shown in the figure.

The language needs improvement to make sure the manuscript clear, unambiguous, and professional. For example, in line 23-24, DEPs (differentially expressed proteins) are comparison between two groups, but the authors only mentioned the germinating group. In line 34, the word “underly” is not professional, consider revision.
Also, minor typos are present in the manuscript. Some examples are: In line 41, a space between “saponins” and “are” is missing; In line 67, there are two periods.

The English language of this manuscript needs improvement to make sure international audience understand it. Some examples are line 45, 102, 268.

Experimental design

"No comment".
The authors of the manuscript entitled “Comparative proteomic analysis between mature and germinating seeds in Paris polyphylla var. yunnanensis” filled the knowledge gap of the difference between proteome and transcriptome of mature and germinating seeds of Paris polyphylla var. yunnanensis. The experiment design is logical and sound.

Validity of the findings

Conclusions are too tedious, try to focus on important results/findings.

---

## Round 0.2 · Major Revisions

Both reviewers indicated that your manuscript has numerous linguistic issues. Therefore, you should contact a professional editor or a proficient English speaker to improve your manuscript. Please note that PeerJ can provide language editing services. Please contact them at [email protected] for pricing (be sure to provide your manuscript number and title).

·

Basic reporting

The authors have addressed almost all my concern. However, the writing needs to be further improved.

Experimental design

The metods are detail described.

Validity of the findings

The conclusions is appropriately

Additional comments

I suggest authors ask a language service.

Reviewer 2 ·

Basic reporting

Even though the authors have revised the manuscript, the language still needs improvement to make sure the manuscript clear, unambiguous and professional. For example, in the abstract, past tense should be used when the authors were describing the method they used. Through the paper, the author used some present perfect that doesn't seem appropriate.

The figures still seem fussy to me. Please provide figures with higher resolution.

Experimental design

no comment

Validity of the findings

no comment

---

## Round 0.3 · Minor Revisions

All remaining issues were addressed and manuscript was amended accordingly. However, it was indicated by the Section Editor that "The authors should submit the raw mass spectrometry data from proteomics to a public repository as well, e.g. at zenodo or at ProteomeXchange/PRIDE data and add to the data availability statement."

Reviewer 2 ·

Basic reporting

The language of the manuscript improved after revision. I don't have other comments.

Experimental design

No comment.

Validity of the findings

No comment.

---

## Round 0.4 · accepted · Accept

The authors uploaded raw data as requested. Revised manuscript is acceptable now.